# Sequence and entropy-based control of complex coacervates

Li-Wei Chang[1], Tyler K. Lytle[2], Mithun Radhakrishna [3], Jason J. Madinya[4], Jon Vélez[1], Charles E. Sing[4] & Sarah L. Perry[1]

Biomacromolecules rely on the precise placement of monomers to encode information for structure, function, and physiology. Efforts to emulate this complexity via the synthetic control of chemical sequence in polymers are finding success; however, there is little understanding of how to translate monomer sequence to physical material properties. Here we establish design rules for implementing this sequence-control in materials known as complex coacervates. These materials are formed by the associative phase separation of oppositely charged polyelectrolytes into polyelectrolyte dense (coacervate) and polyelectrolyte dilute (supernatant) phases. We demonstrate that patterns of charges can profoundly affect the charge–charge associations that drive this process. Furthermore, we establish the physical origin of this pattern-dependent interaction: there is a nuanced combination of structural changes in the dense coacervate phase and a 1D confinement of counterions due to patterns along polymers in the supernatant phase.

[1] University of Massachusetts Amherst, Department of Chemical Engineering, Amherst, MA 01003, USA. [2] University of Illinois at Urbana-Champaign, Department of Chemistry, Urbana, IL 61801, USA. [3] Indian Institute of Technology Gandhinagar, Department of Chemical Engineering, Gandhinagar, 382355, India. [4] University of Illinois at Urbana-Champaign, Department of Chemical and Biomolecular Engineering, Urbana, IL 61801, USA. L.-W. Chang and T.K. Lytle contributed equally to this work. Correspondence and requests for materials should be addressed to C.E.S. (email: cesing@illinois.edu) or to S.L.P. (email: perrys@engin.umass.edu)

Polymer properties follow primarily from their one-dimensional nature, with their length distinguishing them from other soft materials. This length is due to the end-to-end connection of monomer units; the precise sequence of these monomers is capable of encoding information along the backbone[1, 2]. However, interactions between these long chains are typically described in synthetic polymers by coarse-grained effective interactions between immediate neighboring molecules[3]. Polymer physics relies on the use of the these interactions, described by a parameter $\chi$, which has its origins in average, pairwise, short-range interactions[3, 4]. Biological materials, however, use a richer array of polymer–polymer interactions where this sort of 'averaging' may obscure relevant physical properties[5] and limit our ability to understand the complicated biological structure–function relationships encoded at the molecular level. The use of charge in sequence-controlled biopolymers is ubiquitous[6–8]. For example, charge sequence is shown to dictate the conformational behavior of intrinsically disordered proteins (IDPs)[9], and theoretical work has similarly connected IDP sequence to charge-driven phase separation[10]. Sequence is correspondingly a key aspect of intracellular compartmentalization via membrane-less organelles[6].

While solid-phase synthesis methods[11, 12] have long been used to prepare sequence-controlled polymers, recent advances in synthetic polymer chemistry have expanded the palette of sequence-defined polymerization methods[1, 2, 13–15]. For instance, advances in chemical synthesis have enabled the evaluation of precise charge spacing effects in ionomers[16, 17]. However, a general understanding of the physics of sequence-defined polymer materials remains underdeveloped.

Initial efforts have begun to elucidate how monomer sequence physically influences polymer material properties. In particular, the continuum of behaviors between block and random co-polymers has been probed in terms of equilibrium properties (e.g., phase behavior[18, 19], compatibilization[20]) using coarse-grained modeling and theory. These works consider portions of a vast sequence parameter space, using monomer sequence correlations (i.e., blockiness)[18, 19], sophisticated machine learning methods[20], or sequence gradients[21]. These situations focus on short-range dispersive interactions, where monomers interact primarily with their immediate neighbors. Charge interactions differ from short-range interactions, leading to different types of design rules; this difference can be tied to both the long-range nature of electrostatic interactions, and the complementarity between positive and negative charges suppressing like interactions and promoting partner interactions.

In this article, we demonstrate that sequence specificity of charged monomers can be used to precisely control the self-assembly and thermodynamics of a class of materials known as complex coacervates[22, 23]. Charge-based sequence control allows for dramatic modulation of polymer–polymer interaction strengths without changing the overall monomer composition. We experimentally and computationally demonstrate the effects of charge patterning, and establish the physical picture and design rules necessary to show why charge patterning has such a profound effect on coacervate phase behavior.

## Results

**Oppositely charged polymers drive self-assembly.** Oppositely charged polyelectrolytes can undergo associative phase separation in an aqueous solution, forming a polymer-dense coacervate phase and a polymer-dilute supernatant phase[22, 23]. This process is known as complex coacervation, which broadly describes any liquid–liquid phase separation due to oppositely charged species. Recent experimental work into the fundamental physics of polymer–polymer coacervation[24–27] is motivated by efforts to use this motif to drive self-assembly[28–33]. Similarly, advances in coacervate theory have led to a range of field theoretic[34–37] and phenomenological[38–41] models of coacervation[42, 43].

Figure 1a schematically illustrates a standard complex coacervate phase diagram, in the space spanned by salt concentration $c_S$ and polymer concentration $c_P$ (Methods section). At low salt and polymer concentrations, in the coexistence region (2Φ) underneath the binodal curve, the system spontaneously undergoes a phase separation into the high-$c_P$ coacervate phase and the low-$c_P$ supernatant phase. The coacervate and supernatant states are connected along a tie line, which is sloped to denote a difference in $c_S$ between the two phases. Beyond the coexistence region, the system becomes completely miscible. Previous work has demonstrated that this phase diagram is extremely sensitive to molecular-level structure[44, 45]. Changes in bond length and charge size can drastically expand or shrink the coexistence region, reflecting differences in local charge correlations that arise between the highly connected, oppositely charged polyelectrolytes[45]. However, it is difficult to experimentally demonstrate these effects in a controllable fashion. Instead, changing charge monomer

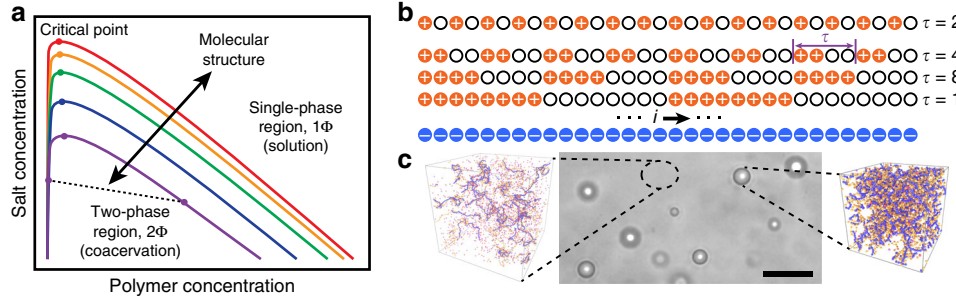

**Fig. 1** Molecular structure and sequence affects charge-driven phase separation. **a** Qualitative sketch of a typical phase diagram of complex coacervate-forming polyelectrolytes. Coacervation occurs at low salt and polymer concentrations, where oppositely charged polyelectrolytes undergo a liquid-liquid phase separation into polymer dense (coacervate) and polymer-dilute (supernatant) phases. The different curves qualitatively represent how the immiscible region changes with different molecular features (charge monomer sequence, spacing, ion size, degree of polymerization, valency, etc.). **b** We show that charge monomer sequence is a molecular feature, which can be used to tune coacervation behavior. This simulation and experimental result is based on coacervation between a homopolyanion and a series of model, sequence-defined polycations with half of their monomers charged. These polycations are characterized by the periodic repeat of the monomer sequence, $\tau$. **c** Coacervation is experimentally observed as droplets of a polymer-dense 'coacervate' dispersed in a polymer-dilute 'supernatant' phase. Simulation images correspond to conditions (salt concentration, 25 mM and $\tau = 2$) shown in Fig. 2. Scale bar is 25 μm

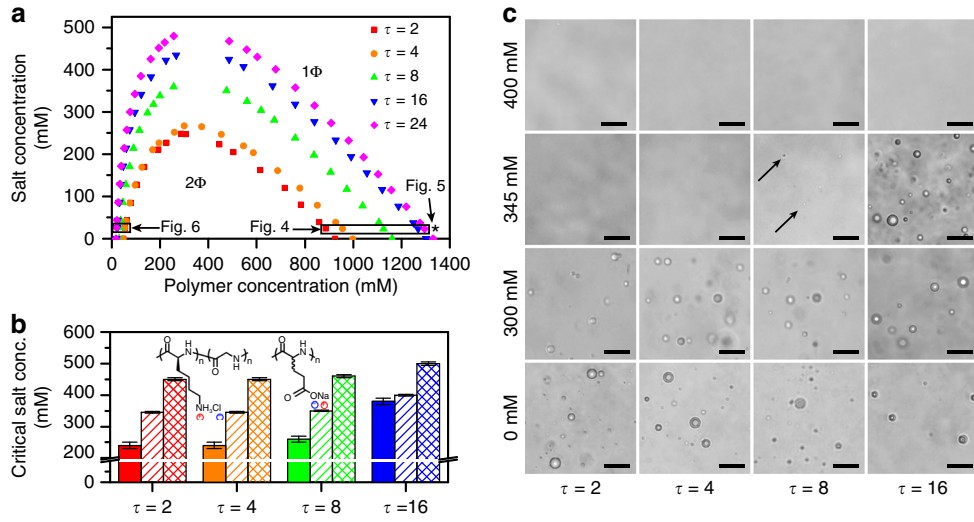

**Fig. 2** Coacervate phase behavior is affected by charge sequence in both simulation and experiment. **a** Simulations demonstrate that the size of the coexistence region 2Φ increases with $\tau$. Simulation conditions for Figs. 4–6 are specified by asterisks/boxes, which denote points along the binodal curves at 25 mM NaCl. These points are considered, because the salt concentration values correspond to those used for isothermal titration calorimetry. **b** The experimental critical salt concentration (CSC) for sequence-defined coacervates at a variety of total charged monomer concentrations (solid 1 mM, stripes 5 mM, crosshatch 50 mM), plotted as a function their periodic block size ($\tau = 2$ to $\tau = 24$). Increasing $\tau$ leads to a marked increase in the CSC, qualitatively changing by as much as 50–150 mM, consistent with simulations in **a**. Error bars reflect the intervals between samples in these experiments. **c** A selection of optical micrographs corresponding to the data in **a**, highlighting that the region of coacervation increases with $\tau$. Arrows indicate the presence of tiny coacervate drops. Scale bars are 25 μm

sequence provides both a way to experimentally observe the interplay between electrostatics and molecular structure, and enables the sequence-driven design of coacervate-based materials.

**Tuning molecular interactions via patterning.** We use the 1D pattern of charged monomers along a polymer backbone to controllably tune the local arrangement of charges, and thus the strength of charge interactions between coacervate-forming chains. Experimentally, we consider coacervation between an anionic homopolymer of poly(glutamate) and sequence-specific cationic co-polymers of poly(glycine-co-lysine). These are prepared in aqueous solution with NaCl salt at pH 7.0. All polymers have the same degree of polymerization $N = 50$; because the sequence-specific polycations have a charge monomer fraction of $f = 0.5$, there are twice as many polycation molecules as polyanion molecules to balance the number of charges on these species.

In simulation, a restricted primitive model (RPM) representation is used for the polyelectrolytes and salt[46]. RPM coarse-grains atomistic features of charged systems, representing each species $i$ as beads (salt) or connected beads (polymer) with hard core potentials of radius $a_i$ and a charge of $z_i$. Water is a continuum solvent with dielectric constant $\epsilon = 78.5\epsilon_0$. There are well-established limitations to RPM, which does not account for Hofmeister effects or water structure[47]; however, RPM still accounts for the major trends seen in this paper. See the Supplementary Methods for a detailed description of the model. Figure 1 demonstrates our scheme for the homopolyanion and sequence-specific copolycation. The homopolyanion and copolycation both consists of chains of $N = 48$; similar to experiment, twice as many polycations are present per polyanion. Copolycation sequences for both simulation and experiment are defined by their periodicity $\tau$. A copolycation with a sequence that alternates between charged and neutral monomers would have a value $\tau = 2$, while a copolycation that has eight charged monomers followed by a block of eight neutral monomers has a

periodicity $\tau = 16$ (Fig. 1b). For all sequences, the copolycation has the same number of charged and neutral monomers.

Figure 2a shows the coacervation phase diagrams for a series of patterned copolycations interacting with unpatterned homopolyanions, calculated from simulation. These phase diagrams exhibit a drastic, monotonic increase in the size of the coexistence region. In fact, the critical salt concentration (CSC) nearly doubles from $\tau = 2$ to $\tau = 24$.

Changes in the size of the coexistance region determined from simulation are reflected experimentally by trends in CSC as a function of $\tau$ at a constant polymer concentration, (Fig. 2b, c) with qualitative agreement. While matching between the simulation and experimental results is in part dependent on the choice of simulation parameters such as bead radii, the trend observed here persists regardless of the choice of reasonable parameterization values. We note that this effect persists even when the solvent is changed, with a similar effect of $\tau$ on the CSC in a water/isopropanol solvent mixture (Supplementary Fig. 1).

**Thermodynamics of sequence-defined coacervation.** We use isothermal titration calorimetry (ITC) as a tool to experimentally probe the thermodynamics of complex coacervation (Fig. 3a)[48]. A two-step model of coacervation enables analysis of ITC data and its separation into entropic and enthalpic contributions; 'ion pairing' between oppositely charged polymers is followed by a 'coacervation' step that results in phase separation (fit to raw data shown in Fig. 3a inset)[48].

ITC measurements show a small, positive enthalpic contribution to coacervation, consistent with the results of previous investigations (Fig. 3; Supplementary Methods)[40, 48]. Variations between different sequences are difficult to resolve due to the small magnitude of this term. In contrast, and as expected, entropy is the primary driving force for coacervation[38, 40, 48]. Calculated values for $-T\Delta S$ are both negative and an order of magnitude larger than the observed enthalpies. Furthermore, the entropic driving force for coacervation increases with increasing

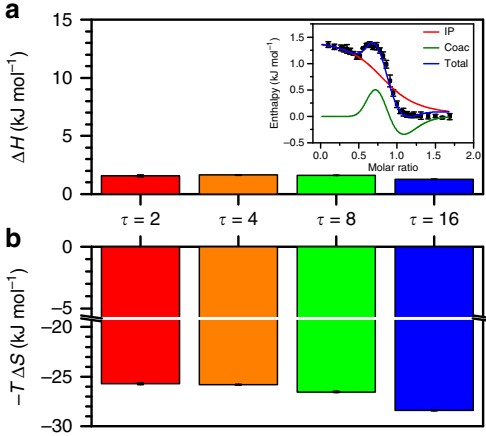

**Fig. 3** Isothermal titration calorimetry (ITC) shows that sequence effects in coacervation are entropically driven. **a** The enthalpic contribution to coacervation as a function of $\tau$ is small, positive, and does not show significant differences between sequences. Isothermal titration calorimatry captures this thermodynamic value via a fit to an established two-step coacervation model (inset) that distinguishes between enthalpic contributions from ion pairing (IP) and coacervation (Coac) steps[48]. **b** The entropic contribution to the coacervation free energy is large, negative, and attributed to counterion release. Clear differences are observed as a function of $\tau$, with an increasing entropic driving force with increasing blockiness (larger $\tau$)

$\tau$, concomitant with the changes in the width of the coexistence region and the CSC observed in simulation and experiment. Furthermore, the magnitude of the entropic differences are significant, spanning ~3 kJ mol$^{-1}$. This is on the order of thermal energy (~1–2$k_\mathrm{B}T$), which can significantly compete against the translational entropy of the polymer chains. This is conceptually consistent with the observed differences in the phase behavior of the different sequences.

**Correlations and sequence alignment in coacervation.** We use simulation to understand the role of charge sequence in determining molecular structure of the coacervate phase. We first consider pair correlations under conditions of constant salt concentration (25 mM) corresponding to the high polymer concentration points on the binodal curves (boxed points in Fig. 2a). These polymer concentrations are relevant for the thermodynamics of coacervation, because they are obtained when coacervation occurs within the two-phase region. The polymer concentration thus depends on the sequence due to the changes in the phase diagram with $\tau$. We focus on the polyanion–polycation correlations $g_{P+/P-}$ (r) shown in Fig. 4a. Peaks corresponding to chain connected structure are seen[44, 45], but there is no clear trend as $\tau$ is changed. This is consistent with a calcuation of the energy of coacervation in Fig. 4b, calculated from $U_i = 2\pi \sum_j \rho_j \int_0^\infty \mathrm{d}r r^2 v_{ij}(r) g_{ij}(r)$[46]. This summates the energy that a species $i$ 'feels' due to contributions from all other species $j$, each with a number density $\rho_j$ and an interaction with $i$ via a pair potential $v_{ij}$[46]. The overall change in energy $\Delta U = \Delta U_{P+} + \Delta U_{P-}$ for coacervation matches with experimental ITC measurements, demonstrating only a small, positive increase that does not depend on $\tau$. This is consistent with the experimental observation that enthalpic effects tend to not dominate the coacervation process[40, 48].

While the coacervate process is not strongly affected by enthalpic effects in coacervation, the structure of coacervates still exhibits non-trivial correlations associated with the monomer sequences. We use a second comparison where dense phases

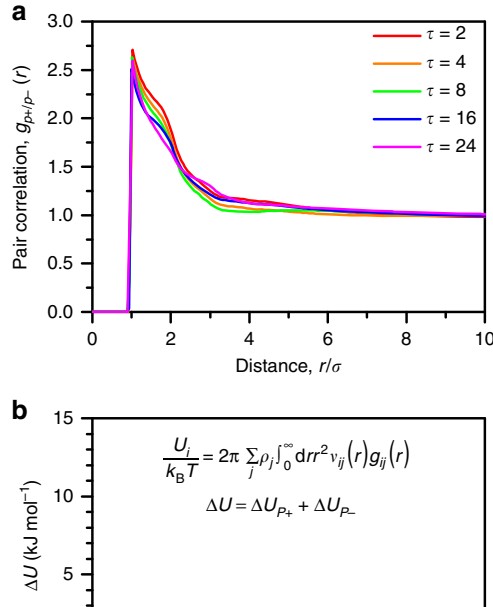

**Fig. 4** Phase separating coacervate structure and energy shows no significant sequence effect. **a** Polycation/polyanion pair correlation function for the coacervate phase at various $\tau$ (boxed points in Fig. 2a). Correlations do not show strong dependence on $\tau$. **b** Calculation of the change in electrostatic energy for the polycation (from g(r) such as in **a**) show small, positive increases in energy during coacervation. This is qualitatively consistent with experimental data in Fig. 3

(denoted with an asterisk in Fig. 2a) for all values of $\tau$ are considered at the same polymer and salt concentrations. This permits a direct comparison between systems with exactly the same components—such as the number of charged/neutral monomers and salt ions—with the only change being the order in which the monomers are connected. Pair correlations $g_{P+/P-}$ (r) are shown in Fig. 5a for all values of $\tau$, demonstrating a distinct change in the second peak adjacent to the initial polyanion/ polycation pair.

The change in this peak can be interpreted through the use of a set of along-the-chain correlation functions $C_1(\Delta s)$ and $C_2(\Delta s)$, which are a function of the distance along a chain contour $\Delta s$ described by the index $s$. We show schematics in Fig. 5b and provide rigorous definitions in Supplementary Note 1. Both functions start with a pair of polycation/polyanion charges that are within a cutoff radius $r_\mathrm{C}$ from each other, and measure conditional probabilities for two monomers that are $\Delta s$ monomers away from original pair. $C_1(\Delta s)$ is the probability that these two new monomers are within $r_\mathrm{C}$ from each other given that they are both charged, while $C_2(\Delta s)$ is the probability that these two new monomers are both charged given they are within $r_\mathrm{C}$ from each other. Conceptually, $C_1$ is a measure of the contour length over which two nearby chains of opposite charge remain aligned, which we call a looping correlation. To contrast, $C_2$ is a measure of how much the charged monomers on the patterned chain prefer to be along segments aligned with the opposite polyelectrolyte, which we call a 'sequence alignment' correlation.

$C_1(\Delta s)$ shows a decrease in looping potential with increasing distance along the chain and very little dependence on the value of $\tau$ (Fig. 5c). This indicates that neighboring chains align for approximately the same number of monomers regardless of sequence. A larger correlation effect is apparent in $C_2$ (Fig. 5d).

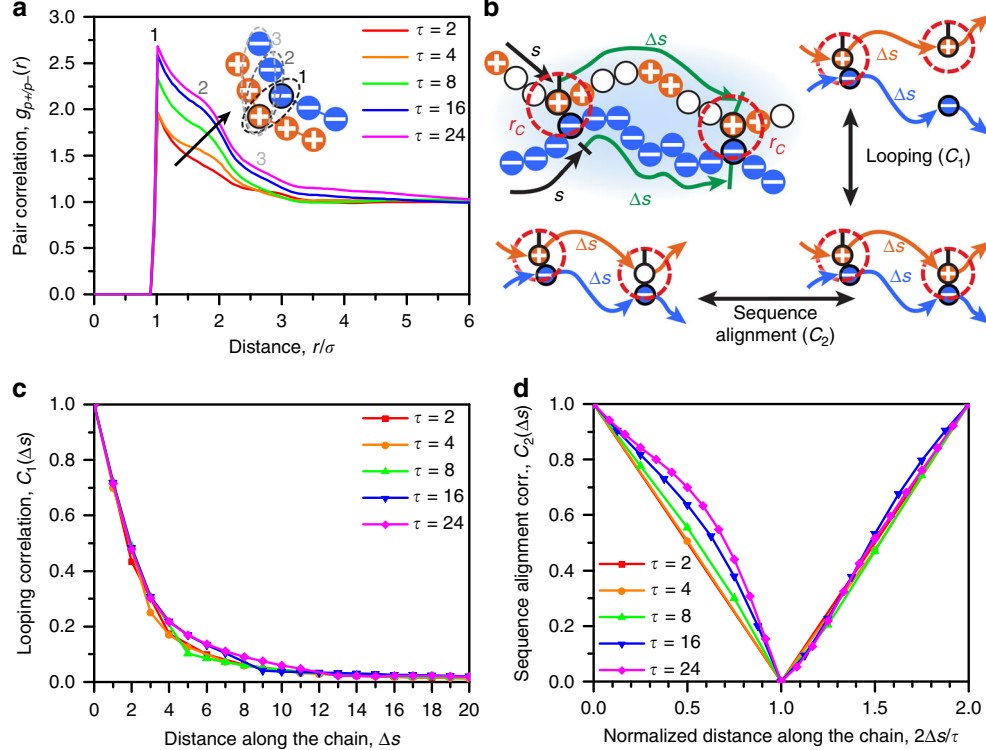

**Fig. 5** Blocky sequences exhibit strong charge correlations due to sequence alignment at the same concentration. **a** Polycation/polyanion pair correlations for the dense phase at a single salt/polymer concentration denoted with an asterisk in Fig. 2a. When species concentrations are kept constant, there is a clear increase in polyelectrolyte correlations. **b** We use a set of pair correlations that capture the extent that two nearby chains interact; we follow their contour $s$ and check for both spatial proximity within a cutoff $r_C$ and monomer charge. $C_1$ determines the probability that charged monomers separated along their respective contours $\Delta s$ loop. $C_2$ determines the probability that looped monomers are both charged. **c** Spatial looping correlations are measured by $C_1$, which demonstrates negligible differences between different values of $\tau$. However, there is a tendency for interacting polyelectrolytes to feature runs of charged monomers, whose sequence alignment is quantified by $C_2$ **d**. We attribute pair correlations in **a** to this effect

Here, the abscissa ($\Delta s$) has been normalized by $\tau/2$ in order to highlight the primary difference between values of $\tau$, which is that the probability of finding another charged monomer after a shift of $\Delta s$ initially decreases much more quickly with small values of $\tau$. In the extreme, for $\tau = 2$, there is by definition no chance of finding a charged monomer for $\Delta s = 1$. To contrast, the likelihood of finding an adjacent charged monomer is very high for large $\tau$, due to the blockier monomer sequence. Beyond this primary probabilistic effect, which is captured by the normalization of $\Delta s$, larger values of $\tau$ still show a slower $C_2$ decay. We attribute this secondary effect to a preference for aligned chain segments to include the charged portion of the patterns. Both of these behaviors are due to the electrostatic benefit of aligning charged monomer sequences, such that opposite charges are in close proximity.

These structural changes at the molecular level do not directly influence the macroscopic thermodynamics of coacervation, as evidenced by the small and $\tau$-independent values of $\Delta U$. Instead, $C_2$ shows that opposite polyelectrolytes tend to align, which entropically confines polyelectrolyte chains in the coacervate phase. This entropic effect is best seen through the lens of counterion release, and is the main driving force for sequence-dependence in coacervation.

**Tuning the entropy of counterion release**. The large entropy change upon coacervation observed in ITC is consistent with traditional counterion release arguments for coacervation[38, 49]. In the dilute phase, counterions condense along the backbone of a highly charged polyelectrolyte to decrease the local electrostatic

energy[49]. This counterion condensation occurs at the expense of the counterion translational entropy. During coacervation, oppositely charged polymers can condense upon each other, similarly lowering the local electrostatic energy. Meanwhile, the previously condensed counterions regain their translational entropy[38, 41, 49, 50]. We use a modified version of this counterion release argument to explain how $\tau$ can strongly affect coacervation phase behavior.

We use simulation to characterize counterion condensation in the dilute phase. We use a method developed by Liu and Muthukumar[51], where condensed counterions are located within a cutoff radius $r_{CC}$ from any monomer along a dilute chain. Each condensed counterion is assigned to its nearest monomer, such that each monomer $i$ has an average number $\langle n_i \rangle$ of counterions condensed (Fig. 6b). The smaller, neutral monomers have a larger accessible counterion volume with this method. A number is therefore defined for each bead using the condensed counterions $\langle n_i^0 \rangle$ for an uncharged chain. The ratio $\langle n_i \rangle / \langle n_i^0 \rangle$ thus gives a normalized measure of the condensed counterions. We relate this ratio to an effective energy $\epsilon_i = -k_B T \ln \left( \langle n_i \rangle / \langle n_i^0 \rangle \right)$ in a one-dimensional adsorption model that is suited to the high charge densities considered in this work (see Supplementary Note 2). The quantity $\ln \left( \langle n_i \rangle / \langle n_i^0 \rangle \right)$ is plotted as a function of monomer index $i$ for a number of different values of $\tau$ (Fig. 6a). The distribution of counterions along the backbone varies greatly, with low $\tau$ polymers showing relatively uniform condensation while high $\tau$ polymers have condensed counterions clustered near the charge blocks (Fig. 6c).

To evaluate the effect of this distribution of condensed counterions on the counterion release entropy, we use an

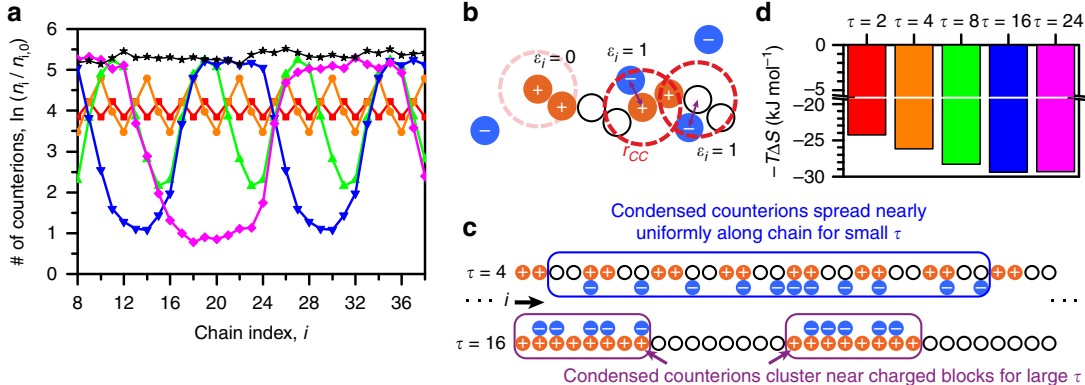

**Fig. 6** Charge sequence effects in coacervation can be explained by 1D counterion confinement entropy. **a** The number of counterions $n_i$ condensed as a function of chain index $i$, relative to the counterions present near an uncharged chain, $n_{i,0}$. Salt concentration is 25 mM, at boxed supernatant points in Fig. 2a. The value $\ln(n_i/n_{i,0})$ is related to an effective binding energy used in a 1D adsorption model. Colors same as Fig. 2a and **d**, black curve for homopolyanion. **b** The criterion for a condensed counterion is one that is within $r_{CC}$ of a polyelectrolyte charge; it is 'condensed' along the nearest polymer bead of index $i$. **c** Conceptual schematic demonstrating the origin of the charge sequence effect on coacervation. Condensed counterions are uniformly distributed along polyelectrolyte chains with low $\tau$, however at high $\tau$ these condensed counterions are confined along-the-chain contour near the charged blocks. This additional confinement increases the entropic driving force for counterion release. **d** This 1D confinement is reflected in the entropic contribution to the free energy, $-T\Delta S$, as calculated from the 1D adsorption model and in near-quantitative matching with ITC data (Fig. 3b)

expression for the entropy calculated from the same one-dimensional adsorption model (energies normalized by $k_B T$ denoted with a tilde):

$$\frac{S}{k_B} = \sum_i \left[ \ln\left(1 + e^{-(\tilde{\epsilon}_i - \tilde{\mu})}\right) + \tilde{\epsilon}_i \left(\frac{e^{-(\tilde{\epsilon}_i - \tilde{\mu})}}{1 + e^{-(\tilde{\epsilon}_i - \tilde{\mu})}}\right) \right]. \quad (1)$$

In this model, simulation data serves as the primary input of $\tilde{\epsilon}_i$, while the external chemical potential $\tilde{\mu}$ is set at a constant value for all $\tau$ and $i$ for a given salt concentration.

Using a single value of $\tilde{\mu}$, we obtain values for the entropic contribution to coacervation in near-quantitative agreement with ITC data (Fig. 6d). Thus, accounting for the distribution of counterions condensed onto individual polyelectrolytes in the supernatant phase yields a prediction for the sequence-dependence of coacervation. This is a one-dimensional confinement effect. Low-$\tau$ systems show an even distribution of condensed counterions along the length of the polyelectrolyte chain (Fig. 6c; $\tau = 4$). However, as $\tau$ is increased, the counterions are increasingly confined near the charged blocks (Fig. 6c; $\tau = 16$). Counterions that are more confined consequently gain more entropy upon release, leading to the increasingly negative values of $-T\Delta S$ with increasing $\tau$ observed in Figs. 3b and 6d.

## Discussion

We used a combination of experiment, theory, and simulation to demonstrate the profound effect of polyelectrolyte monomer sequence on charge-driven materials structure and thermodynamics. Sequence-defined polypeptides were used to evaluate this monomer sequence effect, demonstrating qualitative matching with simulation. This sequence effect is due to differences in entropic confinement of condensed counterions along the polymer, which changes drastically with the blockiness of the sequence. Experimental thermodynamic measurements are consistent with this picture, showing that entropy dominates coacervation while enthalpic contributions are negligible.

We emphasize that this charge patterning effect does not rely on subtle chemical or solvent-specific effects, and trends can be captured using coarse-grained electrostatic models. However, we note that such effects would be important to obtain quantitative predictions. Implications for these charge patterning effects extend from biological polymers to materials design. Sequences

featuring runs of similarly charged macromolecules may provide a way to tune biophysical interactions, with long, charge-dense sequences exhibiting stronger charge interactions than patterns with less-blocky runs of the same charge.

For materials design, charge patterning represents a way to deliberately tune charge interactions in coacervate-driven assembly. This is one way that sequence information may be included into the backbone of a polymer chain that is distinct from, i.e., random copolymerization or block copolymerization. This mechanism is not an averaging of dispersive effects, but rather a precise tuning of the local arrangements of charge. Indeed, by combining with the aforementioned sequence effects we envision a number of sequence-scales that can be used to tune charge-driven assembly. We foresee this as one way to utilize the development of sequence-specific synthesis to reach ever-more complex assemblies.

## Methods

**Coacervation of sequence-controlled peptides**. Polypeptides were prepared via solid-phase synthesis using microwave-enhanced, automated synthesis (Liberty Blue, CEM Corp.) using standard methods[11]. Poly(glutamate) and the poly(glycine-co-lysine) polymer for $\tau = 16$ were synthesized using amino acids of alternating (D and L) chirality to mitigate inter-peptide hydrogen bond formation[52–54]. All other peptides were composed of only L amino acids. See synthesis details in the Supplementary Methods.

Complexation was performed using stoichiometric quantities of positive and negatively charged polypeptides at a total charged residue concentration of 5 mM at pH 7.0 unless otherwise specified. Samples were prepared by first mixing a concentrated solution of NaCl with MilliQ water in a microcentrifuge tube (1.5 mL, Eppendorf), followed by the polyanion. The resulting mixture was then vortexed for 5 s before addition of the polycation to a final volume of 120 µL. The final mixture was vortexed for at least 15 s immediately after the addition of polycation to ensure fast mixing. The effect of salt was examined over the range of 0–520 mM NaCl. All samples were prepared immediately before analysis and studied at room temperature (25 °C). Optical microscopy was used to identify the CSC.

ITC experiments were performed at 25 °C on a MicroCal Auto-iTC200 system (Malvern Instruments, Ltd.) All experiments were performed by injecting a 5 mM solution of the charge-patterned polycation (with respect to the number of lysines) into the sample cell containing 0.625 mM polyanion. Both solutions were prepared at a salt concentration of 25 mM NaCl and pH = 7.0 so as to minimize interference associated with heats of dilution. An initial injection of 0.5 µL was performed, followed by 24 injections of 1 µL each. An injection duration of 2 s followed by a 180 s equilibration time was used. Constant stirring speed was applied at a rate of 1000 rpm. All experiments were performed in triplicate. Analysis of ITC data was performed using the method reported previously[48]. Additional details are available in the Supplementary Methods.

**Monte Carlo informed phase diagram calculations.** The excess free energy, $f_{EXC}$, was determined as a function of polymer and salt concentration from NVT Monte Carlo (MC) simulations[55]. MC simulations were calculated using the system energy:

$$U = \frac{1}{2}\sum_{i,j\neq i}^{N_{tot}}\left[U_{HS}(r_{ij}) + U_{ES}(r_{ij})\right] + \sum_i^n\sum_j\left[U_B(r_{j,j+1}) + U_\theta(\theta_j)\right], \quad (2)$$

where $N_{tot} = N(n_{P_+} + n_{P_-}) + n_+ + n_-$, the total number of beads, $N$ is the degree of polymerization of the chains, $n_{P_+}$ is the number of polycation chains, $n_{P_-}$ is the number of polyanion chains, $r_{ij}$ is the separation between beads $i$ and $j$, $n_+$ is the number of cations, $n_-$ is the number of anions, and $n = n_{P_+} + n_{P_-}$ is the total number of polymer chains. Hard-sphere interactions $U_{HS}(r_{ij})$, electrostatic interactions $U_{ES}(r_{ij})$, a bonding potential $U_B(r_{j,\,j+1})$, and an angle potential, $U_\theta(\theta_j)$, contribute to the overall energy of the system. Excess chemical potentials, $\mu_{EXC,i}$, for each species, $i$, were calculated using Widom insertion. These values were thermodynamically integrated to obtain the excess free energy, $f_{EXC}$[56]. See simulation details in the Supplementary Methods and Supplementary Fig. 7. A Flory–Huggins-like theory was used to calculate binodal curves. The average polymer and salt volume fraction ($\langle\phi_P\rangle, \langle\phi_S\rangle$) determine the system's free energy $\Delta F = F_0 + F_{EXC} - F_{HOM}$, where $F_0$ is the mixing entropy of all species in each phase, and $F_{EXC}$ is the non-ideal contributions determined from MC simulations. $F_{HOM}$ is the free energy of a reference homogeneous phase. $\Delta F$ was minimized to determine $\phi_P$ and $\phi_S$ in both the coacervate and supernatant phases. The resulting phase diagrams include the electrostatic interactions with full correlations via the MC simulations, however the polymer is itself treated at the mean-field level. This limits the accuracy of the model very close to the critical point in the phase diagram, where polymer fluctuations become important, however this will not affect the conclusions of this work. More details about the explicit forms of the various free energies can be found in the Supplementary Methods.

**Data availability.** All data is available from the authors upon reasonable request.

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

## Acknowledgements

C.E.S. acknowledges support from NSF CAREER Award DMR-1654158, J.J.M. acknowledges support from the University of Illinois Graduate College Fellowship. This work used the Extreme Science and Engineering Discovery Environment (XSEDE), which is supported by National Science Foundation Grant ACI-1548562.

## Author contributions

L.-W.C., J.V., and S.L.P. designed, planned, analyzed, and implemented experiments. T.K.L., M.R., J.J.M., and C.E.S. designed, planned, analyzed, and performed simulations. All authors helped with the writing of the manuscript. C.E.S. and S.L.P. designed the overall project.

## Additional information

**Competing interests:** The authors declare no competing financial interests.

