## [Peer Review File · Nature Communications]

Reviewers' comments:

Reviewer #1 (Remarks to the Author):

This is an interesting contribution that blends theory and experiment to connect the impact of sequence patterning on complex coacervation. Overall, the MS is reasonably well written. Those of us who care deeply about complex coacervation and its generalizations are going to be excited about this work. Whether this will be sufficient to engender interest in the broad readership of Nature Communications is a question for which I do not have a persuasive answer. However, given surging interest in the topic of phase separation and the impact of complex coacervation as well as generalizations thereof for cell biology, it is definitely conceivable that this work will command a lot of attention, especially since it is anchored by a well thought out theoretical framework.

Comments to address:

- 1) Figure needs a better discussion and clarification. This is especially true of the caption, which should be fully explanatory of what's being shown in the figure. Is panel (a) the result of a calculation? If so, what do the different colors signify? Does each curve correspond to different values of τ ? These points should be clarified in the figure caption, which is too terse to be understood. As currently presented, Figure 1 seems like a qualitative sketch used to set up the manuscript. If that's the case, it should be spelled out clearly. If, however, panels (a) and (c) are the results of calculations, then this should be spelled out explicitly as well.
- 2) There is an odd phrasing of a sentence on lines 57 - 59 on pg. 3. Is this simply stating that unscreened electrostatic interactions are divergent? The verbiage seems stilted and unduly complicated.
- 3) There is a subtle, albeit important issue to consider with regard to the poly-Glu system. Recent work from the Krishnan lab published in Nature Nanotechnology showed, using single molecule electrometry, that Glu-rich sequences undergo charge regulation even at neutral pH. This comes from upshifted pKa values for specific Glu residues via preferential protonation even at neutral pH. Therefore, the charge of poly-Glu may not be what the authors think it is and it'd be important to account for this aspect in their measurements.
- 4) It would help to plot the width of the two-phase regime as a function of salt concentration to accompany panel (a) in Figure 2. This is akin to analyzing the Ginzburg criterion, which will help delineate the mean-field regime from the putative critical regime and hence allow the reader to connect the measured salt concentrations to the observed behavior in a way that will provide an assessment of the types of fluctuations one should expect.
- 5) There appear to be several issues with Figure 2. In panel (b) it seems logical that we are looking at experimental data, but. However, this doesn't appear to match the results shown in Figure 2c, at least inasmuch as a visual inspection allows us to compare the two sets of results. As an illustration of this mutual incompatibility, for $\tau=2$ at 300 mM NaCl in 2c there are no droplets, but this should be significantly below the CSC. Additionally, it becomes difficult to see how the values in panels 2b and 2c match the simulation results in 2a if experiments are at 5mM total polymer and simulations appear to all begin above 5 mM polymer (based on the X-axis in 2a the simulation binodal appear to start at 20 mM polymer concentration, suggesting that simulations at 5 mM polymer concentration are below the low concentration arm of the binodal at all salt concentrations for all polymers. Perhaps the clearest message is that there is a similar trend in the simulation and experimental results.
- 6) What is correlation between change in coacervation entropies and changes in width of two-phase region/CSC? It appears that the entropies are more like simulations in terms of grouping of different types of polymers with $\tau_2 \approx \tau_4$. The patterning effects as discerned from Figure 3 are a lot weaker than in Figure 2(a) for example. Discussion of this issue would be helpful.

7) There is a formula for the energy of coacervation that is shown on line 140 (pg. 7). The origins of this formula are unclear as are the various terms in the integrand / summand. For example, what is ρ_j , what is $v_{ij}(r)$ and what do the indices i and j signify? Shouldn't the free energy be extractable from the logarithm of $g(r)$ and this can then be used to extract the salt dependence of the free energy as well as the entropy and enthalpy decomposition.

8) The discussion of the pair correlation was somewhat challenging to follow.

It would be very useful to have two separate intuitive diagrams describing what the correlation functions are quantifying. Alternatively, why include $C1(\Delta s)$ at all? It doesn't add anything and somewhat confuses the narrative

$C2(\Delta s)$ can be cast as a fairly intuitive result (charged runs interact with one another leading to local alignment), but the quantification is really nice

The final section is nice, but feels like it's missing a paragraph providing an intuitive explanation for the result

9) Could an alternative explanation for the results observed be described as follows (a) The

entropy associated with condensed counter ions when $\tau=2$ is significantly higher than when τ is big. When $\tau=2$ lots of 'equivalent' monomers (from a counterion condensation perspective. When τ =big there are fewer equivalent monomers. See 5a for a graphical description

(b) As a result, counterion release is much more entropically favorable when τ is large than when τ is small? If this is correct, a facsimile of this narrative is worth including because it is an intuitive result thanks to the calculations and measurements in this MS.

10) As a cautionary note, the context-dependence of the counter-ion condensation must be extremely strongly dependent on the model used

Has an appropriate sensitivity analysis been performed, whereby the impact of dielectric constant and charge density of the counterions have been titrated?

11) From a technical standpoint, $\tau 8$ used alternating D and L amino acids, while all other τ values did not. Why was this racemic mixture used and how is this difference in chirality controlled? Why is this not an issue for other sequences?

12) There's a recent paper that was published as a communication in Biophysical Journal. Please see [http://www.cell.com/biophysj/fulltext/S0006-3495\(17\)30437-X](http://www.cell.com/biophysj/fulltext/S0006-3495(17)30437-X). The title of the paper is: "Phase Separation and Single-Chain Compactness of Charged Disordered Proteins Are Strongly Correlated". This work shows an adaptation of an RPA model to account for sequence encoded charge patterning on the simple coacervation of symmetric polyampholytic sequences comprising Glu and Lys residues that Das & Pappu studied. The findings in the current MS complement the work of Lin and Chan and should be discussed in a revised version of the current MS.

13) Finally, there is one semantic point that needs attention: On pg. 2:

"charge dictates the structure of intrinsically-disordered proteins". IDPs don't adopt singular structures. Therefore, a better description would be "conformational behavior" of intrinsically disordered proteins instead of structure of IDPs.

Reviewer #2 (Remarks to the Author):

This manuscript describes the effects of sequence control on the coacervation of biomacromolecules. Sequence control refers to the patterning or repetition of charged groups along the polymer chain. The main questions explored in this study are how does sequence effect the coacervation phase diagram, and how is coacervation thermodynamically controlled. These questions are explored experimentally and computationally for the case of anionic polyglutamate and anionic poly(glycine-co-lysine). The latter is sequence controlled, in which charged groups

are clustered together at varying frequencies along the polymer chain. Two main conclusions arise from the study. The first is that increasing blockiness (or clustering) of the charged groups expands the coacervation phase separation region in an entropy-driven manner. The second is that small counterion condensation occurs in a 1-D fashion especially as blockiness increases.

First, I will comment on the validity and significance of the conclusions. The first conclusion is not all that surprising, but it is the first (to my knowledge) comprehensive verification of this trend regarding coacervation and patterning in a systematic manner. It is refreshing to see the computational attention to this matter. The sequence-controlled behavior mimics that seen in biological situations and thus speaks to the patterning of biological macromolecules. Further, the fact that this process is entropically driven is also not surprising, as it has been verified by Schlenoff and others for non-sequence controlled cases. However, it remains important that the authors have confirmed this for the sequence controlled case as well. It is my view that the first conclusion is the most important and interesting in a broad perspective. The second conclusion is somewhat shaky because it is my view that atomistic simulations are needed to really stand behind the claim of 1D. Atomistic simulations will show that the ion pairing is quite messy, in which counterions can be shared among multiple neighbors (and also with dynamics considered, the ions may be quite diffuse). This is in contrast to the oversimplified schematic displayed as Figure 5c.

Second, I comment on more specific considerations for this manuscript:

1. Figure 1 and caption. It is difficult to tell what is simulation and what is experiment. Be clear.
2. Figure 2 and caption. Are there error bars for 2b? If no, why not? It is not clear if b is experimental or not.
3. Page 6, last paragraph. Why are the "binodal curves denoted by boxed points in Fig 2a" considered? Seems arbitrary.
4. Figure 3. Q is not explained. I have to read the SI to understand Q. It should be understandable without having to go to the SI.
5. Figure 4 and accompanying text. I am struggling to completely understand the differences between a and c.
6. I would like to see more experiments. Specifically, Figure 2c needs more salt concentrations to feel out the phase space so that it can adequately complement the concentrations. I think 4 salt concentrations for each pattern is not enough to map out an experimental salt-concentration phase diagram. This would strengthen the first conclusion significantly.

Reviewer #3 (Remarks to the Author):

This is a well written and complete study of the complexation of polyelectrolytes with well controlled architecture. The authors provide an extremely relevant combination of experiment, simulation and theory to argue the potential of controlling coacervation. This is exciting as it actually shows how this process can be controlled with only charge placement along the backbone (and without needing to focus on the details of other inter-monomer potentials). In that, the manuscript is relevant to a very wide range of complexation phenomena from biological molecules through to soft materials design. This meets the criteria of nature communications and I encourage publication.

Since this is written for a communications article, the level of detail and amount of material sent to SI is appropriate. For a full article, some of the material would need to be in the body to justify the interpretation of ITC and characterization of the polypeptides.

Fig 4f is rather difficult to interpret (read). With some thought, it should be possible to find a better way to present sequence alignment along a chain. As this field develops, this will be necessary and I do not think this should stop publication, but there must be a better way to present concepts like that for spatially organized systems. A similar sort of discussion arises with Fig 5a as the language of these organized, highly architecturally controlled systems is developed.

A minor point, in line 111 CSC is used before it is defined (on line 113). It is also italicized in some parts of the document and not others.

We thank the reviewers for their careful reading of our manuscript, and for their helpful comments. They have helped us improve this paper, and we are grateful. We address comments below (italicized) with responses (bolded). When appropriate, we describe changes to the text and highlight changes in red in the revised manuscript. We hope that our revisions make this manuscript suitable for publication in Nature Communications.

Reviewer #1 (Remarks to the Author):

This is an interesting contribution that blends theory and experiment to connect the impact of sequence patterning on complex coacervation. Overall, the MS is reasonably well written. Those of us who care deeply about complex coacervation and its generalizations are going to be excited about this work. Whether this will be sufficient to engender interest in the broad readership of Nature Communications is a question for which I do not have a persuasive answer. However, given surging interest in the topic of phase separation and the impact of complex coacervation as well as generalizations thereof for cell biology, it is definitely conceivable that this work will command a lot of attention, especially since it is anchored by a well thought out theoretical framework.

We appreciate the reviewer's positive response to this work.

Comments to address:

1) Figure needs a better discussion and clarification. This is especially true of the caption, which should be fully explanatory of what's being shown in the figure. Is panel (a) the result of a calculation? If so, what do the different colors signify? Does each curve correspond to different values of tau? These points should be clarified in the figure caption, which is too terse to be understood. As currently presented, Figure 1 seems like a qualitative sketch used to set up the manuscript. If that's the case, it should be spelled out clearly. If, however, panels (a) and (c) are the results of calculations, then this should be spelled out explicitly as well.

The caption for Figure 1 was intended to provide a qualitative setup for the paper. Panel (a) is indeed a qualitative sketch, and we have now changed the caption to clarify its role:

“(a) Qualitative sketch of a typical phase diagram of complex coacervate-forming polyelectrolytes. Coacervation occurs at low salt and polymer concentrations, where oppositely-charged polyelectrolytes undergo a liquid-liquid phase separation into polymer dense (coacervate) and polymer dilute (supernatant) phases. The different curves qualitatively represent how the immiscible region changes with molecular features (charge monomer sequence, spacing, ion size, degree of polymerization, valency, etc.). (b) We show that charge monomer sequence is a molecular feature, which can be used to tune coacervation behavior. This simulation and experimental result is based on coacervation between a homopolyanion and a series of model, sequence-defined polycations with half of their monomers charged. These polycations are characterized by the periodic repeat of the monomer sequence, τ . (c) Coacervation is experimentally observed as droplets of a polymer-dense ‘coacervate’ (simulation figure, right) dispersed in a polymer-dilute ‘supernatant’ phase (simulation figure, left). Simulation images correspond to conditions (salt concentration, 25 mM and $\tau = 2$) shown in Fig. 2.”

Here we have also clarified that the simulation figures in (c) correspond to the $\tau = 2$ case in Fig. 2.

2) There is an odd phrasing of a sentence on lines 57 - 59 on pg. 3. Is this simply stating that unscreened electrostatic interactions are divergent? The verbiage seems stilted and unduly complicated.

The intention was to emphasize that the long-range nature of the electrostatic interaction means that the local correlations will play a central role in coacervate thermodynamics. This is tangential to our main point, so we simplify the sentence:

“(1) the long-range nature of electrostatic interactions, and (2) the...”

3) There is a subtle, albeit important issue to consider with regard to the poly-Glu system. Recent work from the Krishnan lab published in Nature Nanotechnology showed, using single molecule electrometry, that Glu-rich sequences undergo charge regulation even at neutral pH. This comes from upshifted pKa values for specific Glu residues via preferential protonation even at neutral pH. Therefore, the charge of poly-Glu may not be what the authors think it is and it'd be important to account for this aspect in their measurements.

We thank the reviewer for drawing our attention to this interesting new report. We agree entirely that charged groups within close proximity will tend to undergo charge regulation over a wide range of pH conditions. The pKa of glutamic acid alone is typically around 4.15, and various reports of poly(glutamic acid) have cited a pKa value around 4.5, which is in agreement with the up-shift reported by Krishnan et al. While we have not performed a direct measurement to determine charge regulation in our system, we did examine the coacervation behavior as a function of the Glu vs. Lys stoichiometry. In these experiments, similar to other reports, we observed that coacervation is sharply peaked at 50/50 Glu vs. Lys stoichiometry. This indirectly suggests that either all Glu residues are charged, or that the charge regulation effects of Glu and Lys cancel out.

4) It would help to plot the width of the two-phase regime as a function of salt concentration to accompany panel (a) in Figure 2. This is akin to analyzing the Ginzburg criterion, which will help delineate the mean-field regime from the putative critical regime and hence allow the reader to connect the measured salt concentrations to the observed behavior in a way that will provide an assessment of the types of fluctuations one should expect.

We appreciate that it would be interesting to know where fluctuations near the critical point go beyond the mean-field regime, however we think that it is beyond the scope of this manuscript to do this question justice. First, we note that the information in the suggested plot can be straightforward to view directly on Fig. 2a. Second, a meaningful evaluation of the Ginzburg criterion would require comparison to the composition fluctuations over a correlation volume. Such a derivation, especially since there are multiple components, would be far too technical for the purposes of this manuscript, which is intended for a broad audience. However, this is an interesting possibility for follow-up work on this topic.

However, in the spirit of this question, we agree that it is important to emphasize that the actual location of the critical point will be slightly different in our mean-field treatment of the polymers compared to a fully fluctuating model. We now add this caveat in the methods section:

“The resulting phase diagrams include the electrostatic interactions with full correlations via the MC simulations, however the polymer is itself treated at the mean-field level. This limits the accuracy of the model very close to the critical point in the phase diagram, where polymer fluctuations become important, however this will not affect the conclusions of this work.”

5) There appear to be several issues with Figure 2. In panel (b) it seems logical that we are looking at experimental data, but. However, this doesn't appear to match the results shown in Figure 2c, at least inasmuch as a visual inspection allows us to compare the two sets of results. As an illustration of this mutual incompatibility, for $\tau=2$ at 300 mM NaCl in 2c there are no droplets, but this should be significantly below the CSC. Additionally, it becomes difficult to see how the values in panels 2b and 2c match the simulation results in 2a if experiments are at 5mM total polymer and simulations appear to all begin above 5 mM polymer (based on the X-axis in 2a the simulation binodal appear to start at 20 mM polymer concentration, suggesting that simulations at 5 mM polymer concentration are below the low

concentration arm of the binodal at all salt concentrations for all polymers. Perhaps the clearest message is that there is a similar trend in the simulation and experimental results.

We first thank the reviewer for pointing out the discrepancy between Fig. 2b and Fig. 2c – this was a mistake when we first put together Fig. 2c, which has now been corrected.

We have incorporated more data into the experiments in Fig. 2 in order to further clarify sequence trends and to address the dependence of the CSC on polymer concentration. We agree with the reviewer that the emphasis should be placed on the trends, since there are quantitative differences (due to the coarse-grained nature of this model). We now emphasize this where relevant in the manuscript.

6) What is correlation between change in coacervation entropies and changes in width of two-phase region/CSC? It appears that the entropies are more like simulations in terms of grouping of different types of polymers with $\tau_2 \approx \tau_4$. The patterning effects as discerned from Figure 3 are a lot weaker than in Figure 2(a) for example. Discussion of this issue would be helpful.

We are focused primarily on the general trends, which we think clearly correlates coacervation entropies (which become larger with τ) to the size of the coexistence region (which also becomes larger with τ). Indeed, the entropies calculated from the simulations in Fig. 2a (seen in the new Fig. 6) show a trend that is similar to Fig. 3.

We think the reviewer implies a conceptual issue that should be addressed in the text, which is that the changes in the entropic contributions seem small compared to the phase diagrams. We now add in language to explain this:

“Furthermore, the magnitude of the entropic differences are significant, spanning ~ 3 kJ/mol. This is on the order of thermal energy ($\sim 1-2k_B T$), which can significantly compete against the translational entropy of the polymer chains. This is conceptually consistent with the observed differences in the phase behavior of the different sequences.”

7) There is a formula for the energy of coacervation that is shown on line 140 (pg. 7). The origins of this formula are unclear as are the various terms in the integrand / summand. For example, what is ρ_j , what is $v_{ij}(r)$ and what do the indices i and j signify? Shouldn't the free energy be extractable from the logarithm of $g(r)$ and this can then used to extract the salt dependence of the free energy as well as the entropy and enthalpy decomposition.

We thank the reviewer for mentioning that we overlooked the explanations of these quantities. This equation calculates the energy between a central molecule of species i and surrounding molecules of species j , given a pair potential $v_{ij}(r)$ between the species. The bulk number density of each species i is ρ_i . We now clarify this equation:

“This summates the energy that a species i ‘feels’ due to contributions from all other species j , each with a number density ρ_j and an interaction with i via a pair potential $v_{ij}(r)$.”

The natural log of $g(r)$ is related to the potential of mean force, which is different from the overall free energy change upon coacervation.

8) The discussion of the pair correlation was somewhat challenging to follow. It would be very useful to have two separate intuitive diagrams describing what the correlation functions are quantifying. Alternatively, why include $C1(\Delta s)$ at all? It doesn't add anything and somewhat confuses the narrative $C2(\Delta s)$ can be cast as a fairly intuitive result (charged runs interact with one another leading to local alignment), but the quantification is really nice. The final section is nice, but feels like it's missing a paragraph providing an intuitive explanation for the result

This was a consistent comment among all the reviewers, and so we have reorganized and elaborated on this aspect of the manuscript to improve on our description of this aspect of the paper. We are combining this response with our response to the other reviewer comments, where there was additional confusion regarding the differences between old Fig. 4a and 4c.

To start, we have separated the old Fig. 4 into two separate figures (the new Figs. 4 and 5). This provides a clear distinction between the discussion of coacervate thermodynamics (enthalpic

contributions, Fig. 4) and the discussion of sequence-driven coacervate structure (Fig. 5). We have adjusted the captions accordingly.

We emphasize this in the text by elaborating more about the new Fig. 4, to draw more of a distinction between it and the new Fig. 5. Specifically, we are more explicit that we are discussing the concentrations at the binodal curves because they are relevant to the thermodynamics of the coacervation process:

“These polymer concentrations are relevant for the thermodynamics of coacervation, because they are obtained when coacervation occurs within the two-phase region. The polymer concentration thus depends on the sequence due to the changes in the phase diagram τ ”

We hope that this provides more contrast between the two sets of $g(r)$.

Regarding the $C_1(\Delta s)$ vs. $C_2(\Delta s)$ functions, we have included more clear schematics to distinguish between them. We think it important to include both, since they represent two contrasting hypotheses for the differences in $g(r)$. $C_1(\Delta s)$ demonstrates that these differences are *not* due to extra chain alignment or more ‘looping’ at high τ , which would otherwise be conceptually sensible. The other measure, $C_2(\Delta s)$, instead demonstrates that the differences are due to *sequence* alignment, where the charges on the patterned polyelectrolyte are more likely to be next to the oppositely-charged polyelectrolyte.

We clarify:

“Conceptually, $C_1(\Delta s)$ is a measure of the contour length over which two nearby chains of opposite charge remain aligned, which we call a ‘looping’ correlation. To contrast, $C_2(\Delta s)$ is a measure of how much the charged monomers on the patterned chain prefer to be along segments aligned with the opposite polyelectrolyte, which we call a ‘sequence alignment’ correlation.”

This is accompanied by additional clarification of $C_1(\Delta s)$ in the next paragraph describing the graphs:

“This indicates that neighboring chains align for approximately the same number of monomers regardless of sequence.”

Finally, we agree with the reviewer that a short, conceptual discussion would be helpful to further explain the old Fig. 5c (new Fig. 6c) and provide general physical context. As τ increases, the counterions are increasingly confined locally along the chain. This confinement represents a decrease in entropy. Consequently, counterion release provides an even greater entropic benefit for polycations with large τ values. We now add to our discussion of this to clarify:

“Counterions that are more confined consequently gain more entropy upon release, leading to the increasingly negative values of $-T\Delta S$ with increasing τ observed in Figs. 3b and 6d.”

We also modify the caption for Fig. 6d:

“This additional confinement increases the entropic driving force for counterion release.”

9) *Could an alternative explanation for the results observed be described as follows (a) The entropy associated with condensed counter ions when $\tau=2$ is significantly higher than when τ is big. When $\tau=2$ lots of ‘equivalent’ monomers (from a counterion condensation perspective. When τ =big there are fewer equivalent monomers. See 5a for a graphical description (b) As a result, counterion release is much more entropically favorable when τ is large than when τ is small? If this is correct, a facsimile of this narrative is worth including because is an intuitive result thanks to the calculations and measurements in this MS.*

This was actually our first hypothesis when studying this system. However, our observation was that the number of condensed counterions on the overall chain was not drastically different with different sequences. This can be (roughly) seen in the new Fig. 6a (old Fig. 5a), where the number of condensed counterions along the chain essentially oscillates around a single value. Thus, the number of counterions released per chain does not change significantly with sequence.

10) *As a cautionary note, the context-dependence of the counter-ion condensation must be extremely strongly dependent on the model used. Has an appropriate sensitivity analysis been performed, whereby*

the impact of dielectric constant and charge density of the counterions have been titrated?

A number of different model choices have been tested in the process of developing our understanding of the connection between counterion condensation and sequence. In particular we have varied the size of the neutral beads, the dielectric constant of the system, and the cutoff radius for what is considered 'condensed'. None of these changes affect the qualitative trends we observe, though may have quantitative effects on the location of the binodal (especially the size of the neutral beads).

We have not systematically tested charge density of the polymer chain, but related work (unpublished) from the Sing group can demonstrate that the number of condensed counterions is not very sensitive to charge spacing, at least in the regimes we are considering. Nevertheless, there is certainly a limit where the charge density would be sufficiently low that the simple one-dimensional adsorption model would no longer suffice - we primarily consider high charge-density chains. We now speak to this limitation:

"...in a one-dimensional adsorption model that is suited to the high charge densities considered in this work."

11) From a technical standpoint, τ used alternating D and L amino acids, while all other τ values did not. Why was this racemic mixture used and how is this difference in chirality controlled? Why is this not an issue for other sequences?

Recent computational and experimental work by the Tirrell and de Pablo groups (Hoffmann *et al.*, *Soft Matter*, 2015 and Pacalin *et al.*, *Eur Phys J*, 2016) have shown that a continuous run of approximately eight homochiral amino acids is required to nucleate the formation of a stable beta strand hydrogen bond network between the backbones of oppositely-charged polypeptides. All coacervates were prepared using an alternating D and L (racemic) polyglutamate to avoid the potential for this issue. The use of alternating chirality amino acids for the $\tau = 16$ peptide (we apologize for the typo indicating $\tau = 8$ in the manuscript) is not expected to affect the results of these experiments (based on results from Perry *et al.*, *Nature Commun*, 2015), and allows for the use of this same material in subsequent experiments with different partnering polyanions (beyond the scope of this work).

*12) There's a recent paper that was published as a communication in *Biophysical Journal*. Please see [http://www.cell.com/biophysj/fulltext/S0006-3495\(17\)30437-X](http://www.cell.com/biophysj/fulltext/S0006-3495(17)30437-X). The title of the paper is: "Phase Separation and Single-Chain Compactness of Charged Disordered Proteins Are Strongly Correlated". This work shows an adaptation of an RPA model to account for sequence encoded charge patterning on the simple coacervation of symmetric polyampholytic sequences comprising Glu and Lys residues that Das & Pappu studied. The findings in the current MS complement the work of Lin and Chan and should be discussed in a revised version of the current MS.*

This is a relevant paper, and we thank the reviewer for bringing it to our attention. It is now cited and discussed briefly:

"...and theoretical work has similarly connected IDP sequence to charge-driven phase separation."

13) Finally, there is one semantic point that needs attention: On pg. 2: "charge dictates the structure of intrinsically-disordered proteins". IDPs don't adopt singular structures. Therefore, a better description would "conformational behavior" of intrinsically disordered proteins instead of structure of IDPs.

This has been changed in the manuscript.

Reviewer #2 (Remarks to the Author):

This manuscript describes the effects of sequence control on the coacervation of biomacromolecules. Sequence control refers to the patterning or repetition of charged groups along the polymer chain. The main questions explored in this study is how does sequence effect the coacervation phase diagram, and how is coacervation thermodynamically controlled. These questions are explored experimentally and

computationally for the case of anionic polyglutamate and anionic poly(glycine-co-lysine). The latter is sequence controlled, in which charged groups are clustered together at varying frequencies along the polymer chain. Two main conclusions arise from the study. The first is that increasing blockiness (or clustering) of the charged groups expands the coacervation phase separation region in an entropy-driven manner. The second is that small counterion condensation occurs in a 1-D fashion especially as blockiness increases.

First, I will comment on the validity and significance of the conclusions. The first conclusion is not all that surprising, but it is the first (to my knowledge) comprehensive verification of this trend regarding coacervation and patterning in a systematic manner. It is refreshing to see the computational attention to this matter. The sequence-controlled behavior mimics that seen in biological situations and thus speaks to the patterning of biological macromolecules. Further, the fact that this process is entropically driven is also not surprising, as it has been verified by Schlenoff and others for non-sequence controlled cases. However, it remains important that the authors have confirmed this for the sequence controlled case as well. It is my view that the first conclusion is the most important and interesting in a broad perspective. The second conclusion is somewhat shaky because it is my view that atomistic simulations are needed to really stand behind the claim of 1D.

Atomistic simulations will show that the ion pairing is quite messy, in which counterions can be shared among multiple neighbors (and also with dynamics considered, the ions may be quite diffuse). This is in contrast to the oversimplified schematic displayed as Figure 5c.

We appreciate the reviewer's overall positive response to our work.

The reviewer is correct that atomistic simulations could provide additional insight. Coarse-grained models such as the one we use require some parameterization to match experiments, and atomistic detail perhaps provides more predictive capabilities. We emphasize, however, that in this work the underlying physical mechanism described is qualitatively consistent with experimental results (ITC thermodynamic results, phase behavior) without the need for atomistic detail.

To address the more specific point about ion pairing, we point out that one does not need atomistic simulations to observe that ions form a diffuse (but localized) 'cloud' around a given polyelectrolyte – this is also present in our coarse-grained simulations. We stress that our criteria for ion pairing is not intended to represent a specific 'bond' or other exclusive interaction, but rather to get a statistical sense for the environment next to a given polyelectrolyte monomer.

The intention of the original figure Fig. 5c (now Fig. 6c) is to provide a conceptual explanation, and was not meant to convey a 'snapshot' of the simulation. We now clarify this in the caption:

"(c) Conceptual schematic demonstrating the origin of the charge sequence effect on coacervation."

We also note that, in reply to the first reviewer, we have expanded our discussion of the concepts related to this figure. We hope that this provides additional context for this figure, to avoid the impression that this is anything other than a simplified representation.

Second, I comment on more specific considerations for this manuscript:

1. Figure 1 and caption. It is difficult to tell what is simulation and what is experiment. Be clear.

Please see our response to Reviewer 1's first comment.

2. Figure 2 and caption. Are there error bars for 2b? If no, why not? It is not clear if b is experimental or not.

We apologize for the lack of error bars in our initial figure. This has been corrected. Error bars now reflect the intervals over which experiments were performed to identify the critical salt concentration, and this has been noted in the caption:

"Error bars reflect the intervals between samples in these experiments."

3. Page 6, last paragraph. Why are the "binodal curves denoted by boxed points in Fig 2a" considered?

Seems arbitrary.

We chose the boxed points because they correspond to the concentrations used for the isothermal titration calorimetry experiments. We now clarify:

“These points are considered because the salt concentration values correspond to the those used for isothermal titration calorimetry.”

4. Figure 3. Q is not explained. I have to read the SI to understand Q. It should be understandable without having to go to the SI.

We have changed the legend for the Fig. 3 inset to be more explicit.

5. Figure 4 and accompanying text. I am struggling to completely understand the differences between a and c.

We have included our response to this point in the related discussion of Reviewer 1’s comment 8.

6. I would like to see more experiments. Specifically, Figure 2c needs more salt concentrations to feel out the phase space so that it can adequately complement the concentrations. I think 4 salt concentrations for each pattern is not enough to map out an experimental salt-concentration phase diagram. This would strengthen the first conclusion significantly.

We agree entirely that the optical micrographs presented in Figure 2c are inadequate to accurately map out a phase diagram. These images represent a sub-selection chosen merely to highlight the changes in sample appearance as a function of salt concentration. We have modified the text in the caption to clarify this point.

To address the reviewer’s concern about the need for more experimental data, we now include results in Fig. 2b for a number of polymer concentrations.

Reviewer #3 (Remarks to the Author):

This is a well written and complete study of the complexation of polyelectrolytes with well controlled architecture. The authors provide an extremely relevant combination of experiment, simulation and theory to argue the potential of controlling coacervation. This is exciting as it actually shows how this process can be controlled with only charge placement along the backbone (and without needing to focus on the details of other inter-monomer potentials). In that, the manuscript is relevant to a very wide range of complexation phenomena from biological molecules through to soft materials design. This meets the criteria of nature communications and I encourage publication.

Since this is written for a communications article, the level of detail and amount of material sent to SI is appropriate. For a full article, some of the material would need to be in the body to justify the interpretation of ITC and characterization of the polypeptides.

We thank the reviewer for their positive and accurate description of our work.

Fig 4f is rather difficult to interpret (read). With some thought, it should be possible to find a better way to present sequence alignment along a chain. As this field develops, this will be necessary and I do not think this should stop publication, but there must be a better way to present concepts like that for spatially organized systems. A similar sort of discussion arises with Fig 5a as the language of these organized, highly architecturally controlled systems is developed.

The reviewer rightly points out that the language of this field of sequence-defined polymers is still nascent, especially in polymer physics, and we look forward to the ‘vocabulary’ being developed. We have tried to further clarify the meaning of $C_2(\Delta s)$ (see comment 8 from Reviewer 1), and have re-plotted Fig. 4f (now Fig. 5d) with the contour coordinate normalized by $\tau/2$. We think that this makes the plot less busy, and thus easier to read.

We change how we describe this plot to match the new figure:

“Here, the abscissa (Δs) has been normalized by $\tau/2$ in order to highlight the primary difference between values of τ , which is that the probability of finding another charged monomer after a shift of Δs initially decreases much more quickly with small values of τ . In the extreme, for $\tau = 2$, there

is by definition no chance of finding a charged monomer for $\Delta s = 1$. To contrast, the likelihood of finding an adjacent charged monomer is very high for large τ , due to the blockier monomer sequence. Beyond this primary probabilistic effect, which is captured by the normalization of Δs , larger values of τ still show a slower C_2 decay. We attribute this secondary effect to a preference for aligned chain segments to include the charged portion of the patterns. Both of these behaviors are due to the electrostatic benefit of aligning charged monomer sequences, such that opposite charges are in close proximity.”

A minor point, in line 111 CSC is used before it is defined (on line 113). It is also italicized in some parts of the document and not others.

We now define CSC earlier and removed italics.

We once more thank the reviewers, and hope that our work is found suitable for publication in Nature Communications.

REVIEWERS' COMMENTS:

Reviewer #1 (Remarks to the Author):

The revised version has answered all of my questions / concerns. I have no further comments.

Reviewer #2 (Remarks to the Author):

The authors have greatly clarified the manuscript. I have two remaining suggestions:

1. In the caption of Figure 2, it should be made clear that (b) is experimental.
2. In the caption of Figure 3, the meanings of IP and Coac (from the inset of 3a) should be explained.

Reviewer #3 (Remarks to the Author):

The authors have addressed all of my comments. I appreciate the efforts to make the contribution even better.

We once more thank the reviewers for their positive assessments. We hereby respond to the remaining reviewer queries:

Reviewer 2:

1. In the caption of Fig. 2, it should be made clear that (b) is experimental.

We have clarified in the Fig. 2 caption.

2. In the caption of Fig. 3, the meanings of IP and Coac (from the inset of 3a) should be explained.

We have now added the following:

“...that distinguishes between enthalpic contributions from ion pairing (IP) and coacervation (Coac) steps.”

We are grateful for the constructive comments from all of the reviewers.